

SciPost Phys. Lect.Notes 61 (2022)

# Quantum neural network classifiers: A tutorial

**Weikang Li[1⋆], Zhide Lu[1] and Dong-Ling Deng[1,2†]**

**1** Center for Quantum Information, IIIS, Tsinghua University,
Beijing 100084, People's Republic of China
**2** Shanghai Qi Zhi Institute, 41th Floor, AI Tower, No. 701 Yunjin Road,
Xuhui District, Shanghai 200232, China

⋆ lwk20@mails.tsinghua.edu.cn, † dldeng@tsinghua.edu.cn,

## Abstract

Machine learning has achieved dramatic success over the past decade, with applications ranging from face recognition to natural language processing. Meanwhile, rapid progress has been made in the field of quantum computation including developing both powerful quantum algorithms and advanced quantum devices. The interplay between machine learning and quantum physics holds the intriguing potential for bringing practical applications to the modern society. Here, we focus on quantum neural networks in the form of parameterized quantum circuits. We will mainly discuss different structures and encoding strategies of quantum neural networks for supervised learning tasks, and benchmark their performance utilizing Yao.jl, a quantum simulation package written in Julia Language. The codes are efficient, aiming to provide convenience for beginners in scientific works such as developing powerful variational quantum learning models and assisting the corresponding experimental demonstrations.

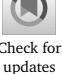

# 1   Introduction

The interplay between machine learning and quantum physics may revolutionize many aspects of our modern society [1–3]. With the recent rise in deep learning, intriguing commercial applications have spread all over the world [4,5]. Remarkably, machine learning methods have cracked a number of problems that are notoriously challenging, such as playing the game of Go with AlphaGo program [6,7] and predicting protein structures with the AlphaFold system [8]. As a powerful method, machine learning may be utilized to solve complex problems in quantum physics. In parallel, over the recent years, quantum-enhanced machine learning models have emerged in various contexts. Notable examples include quantum support vector machines [9,10], quantum generative models [11–13], quantum neural networks [14–17], etc., with some of them holding the potential of providing exponential speedups. In Ref [18], a rigorous quantum speedup is proven in supervised learning tasks assuming some complexity conjectures. With the rapid development of quantum devices, these advanced quantum learning algorithms may also be experimentally demonstrated in the near future.

Artificial neural networks, which can be seen as a highly abstract model of human brains, lie at the heart of modern artificial intelligence [4,5]. Noteworthy ones include feedforward [19,20], convolutional [21], recurrent [22,23], and capsule neural networks [24–28], each of which bears its own special structures and capabilities. More recently, the attention mechanism has been playing an important role in the field of both computer vision [29,30] and natural language processing [31,32], which ignites tremendous interest in exploring powerful and practical deep learning models.

Motivated by the success of classical deep learning as well as advances in quantum computing, quantum neural networks (QNNs), which share similarities with classical neural networks and contain variational parameters, have drawn a wide range of attention [33]. There are multiple reasons to develop the quantum version of neural networks. First, quantum computers hold the potential to outperform classical computers from several aspects: Some quantum Fourier transform based algorithms, such as Shor's factoring algorithm [34], can achieve exponential speedups compared with the best known classical methods; Moreover, some quantum resources such as quantum nonlocality and contextuality are proved to offer unconditional quantum advantages in solving certain computational problems [35,36]. These fascinating results stimulate the exploration of potential advantages in QNN models, especially in the age of big data. Second, when we are trying to learn from a quantum dataset, i.e., a dataset

whose data is generated from a quantum process, instead of from a classical dataset, it would be more natural to use a quantum model the handle the task: Extracting enough information from a quantum state to a classical device would be very challenging when the system scales up, while a QNN model which can handle the data in the exponential-large Hilbert space naturally may provide certain advantages. This point has been explicitly demonstrated in Ref. [15], where the QNN model proposed in this work can recognize quantum states of one-dimensional symmetry-protected topological phases better than existing approaches. Third, from the aspect of effective dimension, which is a property related to a model's generalization performance on new data, there is preliminary evidence showing that quantum neural networks may be able to achieve better effective dimension and faster training than comparable feedforward networks [37].

The early QNN models proposed in the past few years include quantum convolutional neural networks [15], continuous-variable quantum neural networks [38], tree tensor network classifiers, multi-scale entanglement renormalization ansatz classifiers [16], etc. Along this line, various QNN models have been proposed over the past three years [39–47], as well as some theoretical works analyzing QNNs' expressive power [37, 48–57]. For the experimental demonstrations, several QNN models have been implemented experimentally. In Ref. [16], a proof-of-principle QNN classifier is deployed on the ibmqx4 quantum computer to classify the Iris data. In Ref. [58], a QNN classifier is utilized to train and classify some artificially-generated samples on a superconducting quantum processor. With the rapid development of quantum devices, more recently, QNN classifiers are utilized to handle high-dimensional real-life datasets or quantum datasets. In Ref. [59], the authors experimentally demonstrated a quantum convolutional neural network model to identify symmetry-protected topological phases of a spin model on 7-qubit superconducting platforms. In Ref. [60], a quantum neuronal sensing model has been proposed to classify ergodic and localized phases of matter with a 61-qubit superconducting quantum processor. Moreover, in Ref. [61], two QNN classifiers have been experimentally demonstrated for classifying classical and quantum datasets, respectively, where an interleaved QNN classifier is trained on a 36-qubit quantum processor (10 qubits are used) experimentally which turns out to accurately classify 256-dimensional medical images and handwritten digits, and another 10-qubit QNN classifier successfully classifies the quantum states generated by evolving the Néel state with the Aubry-André Hamiltonian. In addition to the fact that QNNs are candidates to be commercially available in the noisy intermediate-scale quantum (NISQ) era [62, 63], it is an interesting point to see that these experimental QNN classifiers introduced above are also suitable to be a measure of progress in quantum techniques during this era.

When designing new QNN models or testing a QNN's performance on a given dataset, an important step is to efficiently simulate the QNN's training dynamics. At the current stage, a number of quantum simulation platforms have been built by institutes and companies worldwide [64–77]. In this paper and the accompanying open-source code, we will utilize Yao.jl [64], a quantum simulation framework written in Julia Language [65], to construct the models introduced and benchmarked in the following sections. For the training process, the automatic differentiation engines ensure the fast calculation of gradients for efficient optimization. To benchmark the performance, we utilize the data from several datasets, e.g., the Fashion MNIST dataset [78], the MNIST handwritten digit dataset [79], and the symmetry-protected topological (SPT) state dataset. The quantum neural network models that we will utilize in the following include amplitude-encoding based QNNs and block-encoding based QNNs. More specifically, amplitude-encoding based QNNs handle data that we have direct access to. For example, if we have a quantum random access memory [80] to fetch the data or the data comes directly from a quantum process, we can assume the data is already prepared at the beginning and use a variational circuit to do the training task. Differently, block-encoding based QNNs

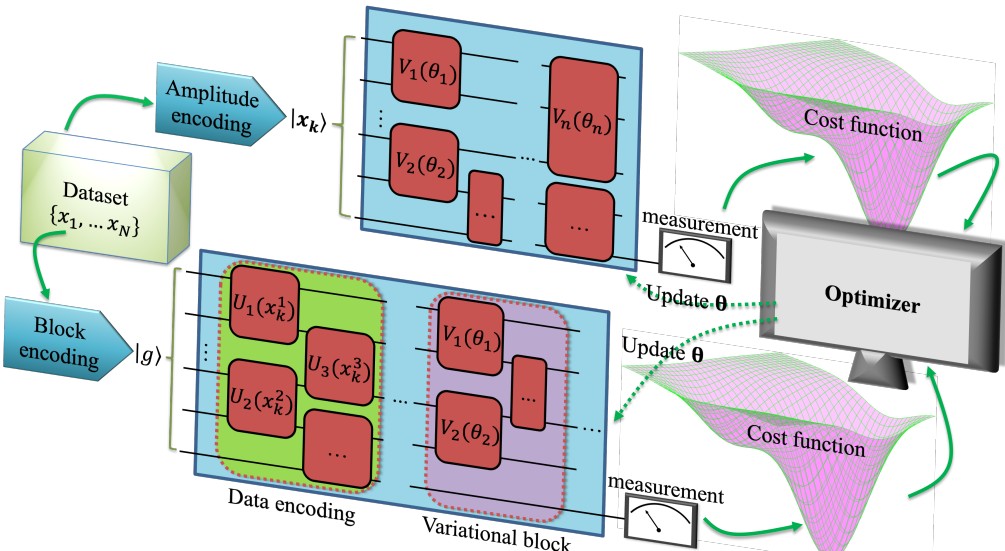

Figure 1: A schematic illustration of quantum neural networks (QNNs) with two data encoding ansatze: amplitude-encoding based QNNs and block-encoding based QNNs. The output states' expectation values of some observables often serve in the cost function to measure the distance between the current and the target predictions, while a classical optimizer is utilized to optimize the QNNs' parameters to minimize the distance.

handle data that we need to encode classically. The classical encoding strategy may seem inefficient, yet it is more practical for experimental demonstrations on the NISQ devices compared with amplitude-encoding based QNNs. One feature of our work is that we mainly focus on the classification of relatively high dimensional data, especially for block-encoding strategies while the present numerical and experimental works are mainly focusing on relatively simple and lower-dimensional datasets. Our benchmarks and open-source repository may provide helpful guidance for future explorations when the datasets scale up.

The sections below are organized as follows: In Section 2, we will recap some basic concepts in quantum computing and give a broader categorization of quantum classifiers. Then, we will review some basic concepts of QNNs, including quantum circuit structures, training strategies, and data encoding ansatze. In Section 3, we will introduce amplitude-encoding based QNNs, which are suitable for situations where we can directly access the quantum data or where we have a quantum random access memory [80] to fetch the data. In Section 4, we will introduce block-encoding based QNNs, which are suitable for situations where we need to encode the classical data into the QNNs. In both Section 3 and Section 4, we will provide codes and performance benchmarks for the models as well as address several caveats. In the last section, we will make a summary and give an outlook for future research. We mention that, due to the explosive growth of quantum neural network models, we are not able to cover all the recent progress. Instead, we choose to focus only on some of the representative models, which depends on the authors' interest and might be highly biased.

## 2 Basic concepts

### 2.1 A recap of quantum computing and quantum classifiers

#### 2.1.1 The basic knowledge of quantum computing

Quantum computing studies the computation model based on the principles of quantum theory, which can harness the features from the quantum world, e.g., superposition and entanglement, to carry out computational tasks. The proposal of quantum computing dates back to 1980 when the quantum Turing machine was introduced by Paul Benioff [81]. A few years later, Richard Feynman proposed to simulate physics with quantum computers, which may avoid the exponential scaling complexity of classical computers [82]. In 1994, an important step in this field was made by Peter Shor, who designed a quantum algorithm that is able to solve the prime factorization problem in polynomial time, i.e., exponentially faster than the best existing classical methods [34]. This algorithm threatens the RSA public-key cryptosystem whose security relies on the hardness of the factorization problem, and further stimulates a wide range of interest and investments into the quantum computing field.

In the quantum computing setting, the basic memory units are quantum bits, also referred to as qubits. To show the difference between bits and qubits, we mention that for a register consisting of $n$ classical bits, there are $2^n$ possible states. For each state, it can be described by a length-$2^n$ vector with exactly one entry being 1 and all the other entries being 0. On the quantum side, the principles of quantum mechanics allow an $n$-qubit quantum state to be described by a $2^n$-dimensional complex vector rather than a single non-zero entry in the classical case, which is usually called the quantum superposition.

With the Dirac notation, we write down the computational basis states in the single-qubit case as

$$|0\rangle = \begin{pmatrix} 1 \\ 0 \end{pmatrix}, \quad |1\rangle = \begin{pmatrix} 0 \\ 1 \end{pmatrix}, \tag{1}$$

and thus a general single-qubit state can be expressed as a linear combination of these two bases:

$$|\psi\rangle = \alpha|0\rangle + \beta|1\rangle = \begin{pmatrix} \alpha \\ \beta \end{pmatrix}, \quad \text{where} \quad |\alpha|^2 + |\beta|^2 = 1. \tag{2}$$

For multi-qubit states, as an example, two quantum states $|\psi\rangle$ and $|\phi\rangle$ can be jointly represented in the tensor product form as $|\psi\rangle \otimes |\phi\rangle$. However, the opposite is not true. Due to the existence of quantum entanglement, a multi-qubit state $|\Psi\rangle$ can not always be decomposed as tensor products of smaller systems, e.g., it is impossible to express the Bell state $|00\rangle + |11\rangle$ as a tensor product of two single-qubit states.

Within the quantum computing framework, it is desirable to design quantum algorithms that can make good use of quantum resources to outperform their classical counterparts. As mentioned above, an important branch of quantum algorithms that runs exponentially faster than the known classical methods is closely related to the quantum Fourier transform. Notable examples include quantum algorithms for solving the prime factorization problem, discrete logarithm problem, order-finding problem, hidden subgroup problem, etc. Besides these, Grover's algorithm [83], i.e., the quantum search algorithm, shows that quantum computers are able to find an input from an unstructured database with quadratic speedups. More recently, Bravyi *et al.* have designed quantum algorithms with shallow quantum circuits that exhibit unconditional quantum speedups in the 2-D hidden linear function problem [35] as well as in some related extended works [36, 84], where the advantages originate from quantum nonlocality and quantum contextuality.

### 2.1.2 A categorization of quantum classifiers

Before we fully get into the introduction of quantum neural network classifiers, it is convenient to provide a broader view of quantum classifiers for better readability. Quantum classifiers are quantum devices that solve classification problems in machine learning, which have been actively studied over the recent years. In general, as a supervised learning task, there should be a training dataset

$$\mathcal{D} = \{(\vec{x}_1, y_1), (\vec{x}_2, y_2), \ldots, (\vec{x}_M, y_M)\}, \tag{3}$$

where $\vec{x}_i$ denotes the vectorized feature of the sample with index $i$, and $y_i$ denotes the corresponding label. A quantum classification model is supposed to learn from this available dataset to automatically attach correct labels to the features with high probability. And after training, the classifier is also supposed to generalize well to some unseen samples.

So far, there are a number of quantum classifiers proposed with some of them implemented experimentally. The categories of these classifiers include quantum nearest-neighbor algorithms, quantum decision tree classifiers, quantum kernel methods, quantum support vector machines, and quantum neural network classifiers. Although the main body of our tutorial focuses on quantum neural network classifiers, we choose to summarize the quantum classifiers mentioned above in Table 1 such that the readers can better learn about this field[1].

Table 1: A list of different quantum classifiers with representative ones exhibited.

| Classifiers | Reference | Brief summary |
|---|---|---|
| QNNA | [85] | Propose a quantum version of the nearest-centroid algorithm that achieves exponential speedups with a QRAM. |
| | [86] | Propose a quantum nearest-neighbor algorithm which achieves quadratic speedup with quantum oracles. |
| QDTC | [87,88] | Introduce the quantum versions of decision tree classifiers. |
| QKM | [58,89] | The first to propose to use quantum computers to evaluate the kernel matrices for supervised learning. |
| | [58,90–92] | Experimental demonstrations of quantum kernel methods. |
| | [18] | The first to prove a rigorous exponential quantum speedup in classification tasks, where the quantum computers are utilized to prepare the kernel matrices. |
| QSVM | [9] | Provide the first discussion about quantum support vector machines with Grover's search algorithm adapted. |
| | [10] | Propose a least-squares quantum support vector machine with the potential of exponential speedups. |
| | [93] | The first work to experimentally implement quantum support vector machines for classifying handwritten digits. |
| QNNC | [15] | Propose the quantum convolutional neural networks for quantum data recognition. |
| | [94] | Propose a quantum analog of classical neurons to build quantum feedforward neural networks. |

## 2.2 The variational circuit structure of QNNs

Quantum neural networks are often presented in the form of parameterized quantum circuits, where the variational parameters can be encoded into the rotation angles of some quantum gates. The basic framework is illustrated in Fig. 1, which mainly consists of the quantum circuit

---

[1]The abbreviations in Table 1 are as follows. QNNA: quantum nearest-neighbor algorithms; QDTC: quantum decision tree classifiers; QKM: quantum kernel methods; QSVM: quantum support vector machines; QNNC: quantum neural network classifiers

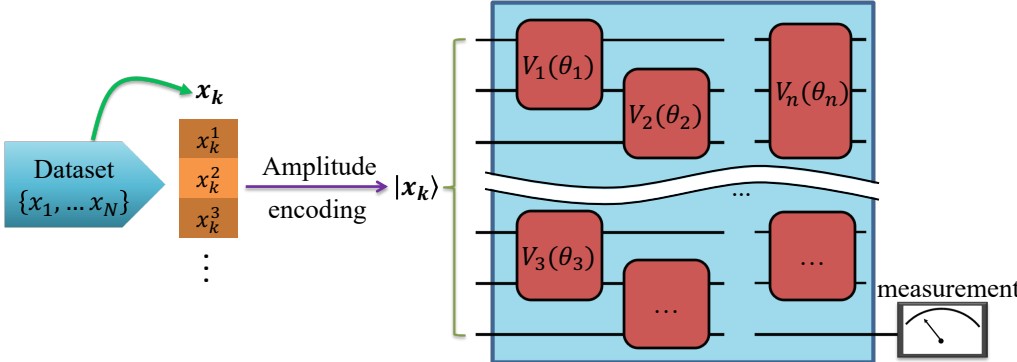

Figure 2: A schematic diagram of amplitude-encoding based QNNs, where the input quantum state can be either from a quantum process (e.g., a quantum system's evolution, quantum state preparation) or from a directly available quantum memory. The followed measurements provide expectation values on some observables, which serve as a classification criterion for making predictions.

ansatz, the cost function ansatz, and the classical optimization strategy. For the basic building blocks, commonly used choices include parameterized single-qubit rotation gates ($R_x(\theta)$, $R_y(\theta)$, and $R_z(\theta)$), Controlled-NOT gates, and Controlled-Z gates, which are illustrated below:

$$\boxed{R_x(\theta)} = e^{-i\frac{\theta}{2}X}, \quad \boxed{R_y(\theta)} = e^{-i\frac{\theta}{2}Y}, \quad \boxed{R_z(\theta)} = e^{-i\frac{\theta}{2}Z},$$

$$= \begin{pmatrix} 1 & 0 & 0 & 0 \\ 0 & 1 & 0 & 0 \\ 0 & 0 & 0 & 1 \\ 0 & 0 & 1 & 0 \end{pmatrix}, \quad \boxed{Z} = \begin{pmatrix} 1 & 0 & 0 & 0 \\ 0 & 1 & 0 & 0 \\ 0 & 0 & 1 & 0 \\ 0 & 0 & 0 & -1 \end{pmatrix}.$$

Here, the rotation angle in a single-qubit rotation gate can be utilized to encode one variational parameter, which can be adjusted during optimization. Besides, there are also a number of blocks that might be convenient for our use, e.g., Controlled-SWAP gates might be experimentally-friendly for demonstrations on superconducting quantum processors. In our numerical simulations, we mainly choose parameterized single-qubit rotation gates ($R_x(\theta)$, $R_y(\theta)$, and $R_z(\theta)$) and Controlled-NOT gates since replacing some of them will not significantly influence the performance. In the following part of this subsection, we mainly introduce two data encoding ansatze: amplitude-encoding ansatz and block-encoding ansatz.

### 2.2.1 Amplitude-encoding ansatz

As the name suggests, amplitude encoding means that the data vector is encoded into the amplitude of the quantum state and then fed to the quantum neural network. In this way, a $2^n$-dimensional vector may be encoded into an $n$-qubit state. The basic structure is illustrated in Fig. 2, where the output can be the expectation values of some measurements and $V_i(\theta_i)$ denotes a variational quantum block with parameter $\theta_i$. It is worthwhile to note that, as indicated in [95, 96], amplitude-encoding based QNNs can be regarded as kernel methods, stressing the importance of the way to encode the data.

### 2.2.2 Block-encoding ansatz

For near-term experimental demonstrations of quantum neural networks, especially for supervised learning tasks, the block-encoding ansatz might be more feasible as there is no need for

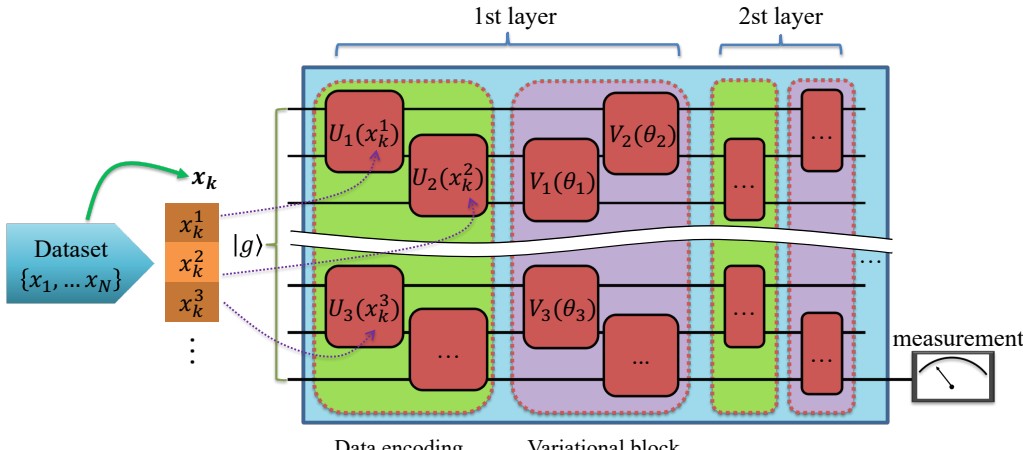

Figure 3: A schematic diagram of block-encoding based QNNs, where the initial quantum state is fixed and the input data requires to be encoded into the QNN circuit classically in a similar way of encoding the variational parameters. The followed measurements provide expectation values on some observables for making the classification decisions.

a quantum random access memory and the data can be directly encoded into the circuit parameters. The basic structure for this ansatz is illustrated in Fig. 3, where the block-encoding strategy can be very flexible with variational quantum blocks $U_i(x_k^i)$ and $V_j(\theta_j)$ encoding the input data and variational parameters, respectively.

## 2.3 Optimization strategies during the training process

### 2.3.1 A general optimization procedure

When we have a quantum neural network model, we wish to train it and apply it to classification tasks. In most cases, we need to first formalize the task to be an optimization problem. How to adapt classification tasks into optimization-based QNN models can be illustrated using the following simple example:

After the data preparation and data encoding procedure, the output state after the QNN is followed by a final measurement $M$. For classification tasks, for simplicity, we can consider a binary classification between two kinds of vectorized digits labeled "cat" and "dog" and assume that the final measurement is applied on the computational basis of a certain qubit. If the input is a "cat", our goal is to maximize the probability of measuring $\hat{\sigma}_z$ with output "1", i.e., maximize $P(|0\rangle)$. Instead, if the input is a "dog", our goal is to maximize the probability of measuring $\hat{\sigma}_z$ with output "−1", i.e., maximize $P(|1\rangle)$. In the prediction phase, we simply compare the probability of different measurement outcomes and assign the label corresponding to the highest probability to the input.

To achieve this goal, we need to optimize the trainable parameters to obtain desirable predictions. To begin with, we generally need to define a cost function to measure the distance between the current output and the target output. Widely used cost functions include the mean square error (MSE) and cross entropy (CE):

$$L_{MSE}(h(\vec{x};\Theta),\mathbf{a}) = \sum_k (a_k - g_k)^2, \tag{4}$$

$$L_{CE}(h(\vec{x};\Theta),\mathbf{a}) = -\sum_k a_k \log g_k, \tag{5}$$

where $\mathbf{a} \equiv (a_1, \cdots, a_m)$ is the label of the input data $\vec{x}$ in the form of one-hot encoding [97], $h$ denotes the hypothesis function determined by the quantum neural network (with parameters collectively denoted by $\Theta$), and $\mathbf{g} \equiv (g_1, \cdots, g_m) = \text{diag}(\rho_{\text{out}})$ presents all the probabilities of the corresponding output categories with $\rho_{\text{out}}$ denoting the output state. Here, $\mathbf{g}$ can be obtained by measuring some qubits on the $Z$-basis. In our tutorial which focuses on binary classifications, we choose to repeatedly measure one qubit on the $Z$-basis to evaluate $\mathbf{g} = (g_1, g_2)$, where the choice of this qubit's index can be flexible.

With a cost function defined, the task of training the quantum neural network to output the correct predictions can be transformed into the task of minimizing the cost function, where gradient-based strategies can be well utilized. In general, after calculating the gradient of the loss function $L$ with respect to the parameters denoted collectively as $\Theta$ at step $t$, the update of parameters can be expressed as

$$\Theta_{t+1} = \Theta_t - \gamma \nabla L(\Theta_t),\tag{6}$$

where $\gamma$ is the learning rate. In practice, gradient methods are often cooperated with optimizers such as Adam [98], to gain higher performance.

Now, we have obtained an overall idea of optimization, while the remaining problem is how to efficiently calculate the gradients. For example, if we take cross entropy as the cost function, the gradient with respect to a particular parameter $\theta$ can be written as

$$\frac{\partial L(h(\vec{x};\Theta),\mathbf{a})}{\partial\theta} = -\sum_k \frac{a_k}{g_k}\frac{\partial g_k}{\partial\theta},\tag{7}$$

where $g_k$ can be seen as an expectation value of an observable, and calculating the gradient of the cost function can be reduced to calculating the gradient of the observables. To address this issue, a number of strategies have been proposed such as the "parameter shift rule" [99–102], quantum natural gradient [103, 104], etc. For a wider-range literature review, readers can refer to review articles in Ref. [14, 33]. In this paper, we choose and explain the basic ideas of parameter shift rules as follows:

For amplitude-encoding ansatz, given an input state $|\psi_x\rangle$, a QNN circuit $U_\Theta$, and an observable $O_k$, the hypothesis function can be written as

$$g_k = h_k(|\psi_x\rangle;\Theta) = \langle x|U_\Theta^\dagger O_k U_\Theta|x\rangle.\tag{8}$$

For block-encoding ansatz, without loss of generality, we assume that the initial state is $|0\rangle$, and the QNN circuit $U_{\Theta,\vec{x}}$ contains both the input data and the trainable parameters. The hypothesis function in this setting can be written as

$$g_k = h_k(|0\rangle,\vec{x};\Theta) = \langle 0|U_{\Theta,\vec{x}}^\dagger O_k U_{\Theta,\vec{x}}|0\rangle.\tag{9}$$

For both the two ansatze, the parameter shift rule tells us that if the parameter $\theta$ is encoded in the form $\mathcal{G}(\theta) = e^{-i\frac{\theta}{2}P_n}$ where $P_n$ belongs to the Pauli group, the derivative of the expectation value with respect to a parameter $\theta$ can be expressed as

$$\frac{\partial g_k}{\partial\theta} = \frac{\partial h_k}{\partial\theta} = \frac{h_k^+ - h_k^-}{2},\tag{10}$$

where $h_k^\pm$ denotes the expectation value of $O_k$ with the parameter $\theta$ being $\theta \pm \frac{\pi}{2}$. In the QNNs presented in this paper, the parameters are encoded in single-qubit Pauli-rotation gates, thus fitting the requirements discussed above. When we look at the expression of the parameter shift rule, we may connect it to a widely-applied approximation method called the finite difference method. In our case, the above derivative can be approximately expressed as

$$\frac{\partial g_k}{\partial\theta} = \frac{\partial h_k}{\partial\theta} \approx \frac{h_k|_{\theta\to\theta+\frac{\Delta\theta}{2}} - h_k|_{\theta\to\theta-\frac{\Delta\theta}{2}}}{\Delta\theta}.\tag{11}$$

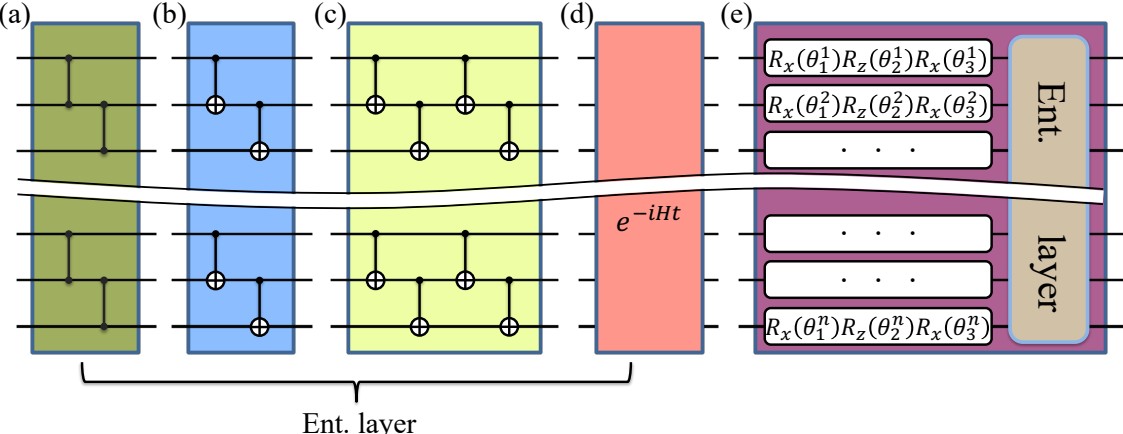

Figure 4: A schematic illustration of the basic building blocks of quantum neural network classifiers. (a) An entangling layer consisting of a single layer of Controlled-Z gates. (b) An entangling layer consisting of a single layer of Controlled-CNOT gates. (c) An entangling layer consisting of two layers of Controlled-CNOT gates. (d) An entangling layer implemented by a many-body Hamiltonian's time evolution. (e) A composite block consisting of three layers of variational single-qubit rotation gates and an entangling layer.

In addition to the fact that the finite difference method is not exact, the unavoidable experimental noise will affect the result with the finite difference method more than that with the parameter shift rule ($\frac{1}{\Delta\theta} \gg \frac{1}{2}$), thus this approximation method is less practical. In the codes attached in this paper, the gradients are calculated by automatic differentiation implemented by Yao.jl. The reason why we do not apply the parameter shift rule is that we wish to make the numerical simulations faster. Automatic differentiation fulfills this goal and meanwhile has analytical precision, thus being the one adapted in our work. When we have a real quantum computer to deploy large-scale quantum neural networks that are hard for classical computers to simulate, methods like the parameter shift rule would be a natural and favorable choice.

### 2.3.2 Effects of finite measurements and experimental noises

In the above discussions about the optimization procedure, the calculation of gradients is closely related to some expectation values obtained from quantum measurements. However, unlike the numerical simulations where we can calculate the accurate expectation values, we can only apply a finite number of measurements to approximate these values with a real quantum computer. Moreover, the unavoidable experimental noises will further drive the outputs away from the accurate values.

First, for the effects of a finite number of measurements, there are a constant number of discrete measurement outcomes with different probabilities. According to the Chernoff bound, $O(\frac{1}{\epsilon^2})$ repeat measurements are needed to achieve the evaluation of an expectation value with an additive error up to $\epsilon$. At the current stage, the superconducting quantum processors can accomplish thousands of single-qubit measurements in several seconds, which is suitable for demonstrating QNNs' learning process [59–61]. Second, since the experimental noises make the evaluations less accurate, the QNN model may converge slower and even get stuck into barren plateaus [105], which might be intractable when the system scales up. To handle this issue, on the one hand, it is necessary to push the experimental limit and improve the gate fidelities. On the other hand, developing QNN models with higher noise robustness is of crucial importance.

## 3 Amplitude-encoding based QNNs

### 3.1 The list of variables and caveats

In the above sections, we have discussed the structures and features of amplitude-encoding based QNNs. Here, we start to numerically explore the performances of amplitude-encoding based QNNs under different hyper-parameter settings. To make the benchmarks more organized, we first list the hyper-parameters in this subsection as well as present some basic codes to numerically build the framework while leaving the benchmarks of these QNNs' performances in the next subsection.

**Entangling layers.** When designing a QNN structure, it is convenient to encode the variational parameters into single-qubit gates. At the same time, we also need entangling layers to connect different qubits. For this purpose, we design two sets of entangling layers. First, we consider entangling layers in a digital circuit setting, e.g., these entangling layers are composed of Controlled-NOT gates or Controlled-Z gates (see Fig. 4(a-c) for illustrations):

```
using Yao, YaoPlots
using Quantum_Neural_Network_Classifiers: ent_cx, ent_cz, params_layer

# number of qubits
num_qubit = 10

# function that creates the entangling layers with Controlled-NOT gates
ent_layer = ent_cx(num_qubit)
# display the entangling layer's structure
YaoPlots.plot(ent_layer)

# function that creates the entangling layers with Controlled-Z gates
ent_layer = ent_cz(num_qubit)
# display the entangling layer's structure
YaoPlots.plot(ent_layer)

# double the entangling layer of Controlled-NOT gates
ent_layer = chain(num_qubit, ent_cx(num_qubit), ent_cx(num_qubit))
# display the entangling layer's structure
YaoPlots.plot(ent_layer)
```

Second, we consider entangling layers in an analog circuit setting, where the time evolution of a given Hamiltonian is utilized as an entangling layer (see Fig. 4(d)):

```
# create entangling layers through Hamiltonian evolutions
t = 1.0 # evolution time

# h denotes the matrix formulation of a predefined Hamiltonian
# the code creating the Hamiltonian may take plenty of space
# thus we leave the Hamiltonian's implementation in the Github repository
@const_gate ent_layer = exp(-im * h * t)
# display the entangling layer's structure
YaoPlots.plot(ent_layer)
```

With these entangling layers, we further define the composite blocks that consist of parameterized single-qubit gates and entangling layers as the building blocks of QNNs from a higher level (see Fig. 4(e)):

```
# given the entangling layer ent_layer
# we define a parameterized layer consisting of 3 layers of single-qubit
# rotation gates: ent_cx(nbit), ent_cz(nbit), and ent_cx(nbit)
parameterized_layer = params_layer(num_qubit)
# display the parameterized layer's structure
YaoPlots.plot(parameterized_layer)

# build a composite block
composite_block(nbit::Int64) = chain(nbit,params_layer(nbit),ent_cx(nbit))
# display the composite block's structure
YaoPlots.plot(composite_block(num_qubit))
```

**Circuit depth.** Intuitively, if a QNN circuit gets deeper, the expressive power should also be higher (before getting saturated). To verify this, we create a set of QNN classifiers with various depths. We denote that, for a QNN classifier with depth $N$, the classifier contains $N$ composite blocks. To implement this in numerical simulations, it is convenient to repeat the composite blocks exhibited above as follows:

```
# set the QNN circuit's depth
depth = 4
# repeat the composite block to reach the target depth
circuit = chain(composite_block(num_qubit) for _ in 1:depth)
dispatch!(circuit,:random)
# display the QNN circuit's structure
YaoPlots.plot(circuit)
```

**A caveat on the global phase.** For amplitude-encoding based QNNs, we would like to mention a caveat concerning the data encoded into the states. It is well-known that if two quantum states only differ by a global phase, they represent the same quantum system. For concreteness, the expectation values of the two states on a given observable will be the same, thus ruling out the possibility of our QNN classifiers distinguishing between them. In practice, the indistinguishability of the "global phase" may also appear in our tasks. For image datasets with objects such as images of handwritten digits or animals, we can convert the images into vectors and normalize them to form quantum datasets, and the QNN will work to handle them in general. Before introducing the caveat, we would like to point out that, for a dataset described by different features (e.g., a mobile phone's length, weight, resolution of the camera, etc), it is convenient to bring down all the features to the same scale before further processings. To achieve this goal, data standardization is often applied such that each rescaled feature value has mean value 0 and standard bias 1. However, for simple datasets with two labels, the standardization operation may mainly create a global phase $\pi$ between the two classes of data. For example, the data in the original first class concentrates to $(1, 1)$ and the data in the original second class concentrates to $(3, 3)$. After standardization, they become $(-1, -1)$ and $(1, 1)$, respectively, which become indistinguishable by our QNN classifiers in the amplitude encoding setting. To overcome this problem, a possible way is to transform the "global phase separation" into "local phase separation": Assume we already have a dataset with two classes of data concentrating on $(-1, -1)$ and $(1, 1)$. We add multiple "1"s to each data vector such that the data becomes $(-1, -1, 1, 1)$ or $(1, 1, 1, 1)$, thus letting the data learnable again by the QNN classifier. We met this problem in practice when handling the Wisconsin Diagnostic Breast Cancer dataset [106] which consists of ten features such as radius, texture, perimeter, etc. The classification is far from ideal when we simply standardize the dataset, and after transforming the "global phase" to "local phase", the training becomes much easier and an accuracy of over 95% is achieved.

---

**Algorithm 1** Amplitude-encoding based quantum neural network classifier

---

**Input:** The untrained model $h$ with variational parameters $\Theta$, the loss function $L$, the training set $\{(|\mathbf{x}_m\rangle, \mathbf{a}_m)\}_{m=1}^n$ with size $n$, the batch size $n_b$, the number of iterations $T$, the learning rate $\epsilon$, and the Adam optimizer $f_{\text{Adam}}$

**Output:** The trained model

  1: Initialization: generate random initial parameters for $\Theta$
  2: **for** $i \in [T]$ **do**
  3:    Randomly choose $n_b$ samples $\{|\mathbf{x}_{(i,1)}\rangle, |\mathbf{x}_{(i,2)}\rangle, ..., |\mathbf{x}_{(i,n_b)}\rangle\}$ from the training set
  4:    Calculate the gradients of $L$ with respect to the parameters $\Theta$, and take the average value over the training batch $\mathbf{G} \leftarrow \frac{1}{n_b}\Sigma_{k=1}^{n_b}\nabla L(h(|\mathbf{x}_{(i,k)}\rangle; \Theta), \mathbf{a}_{(i,k)})$
  5:    Updates: $\Theta \leftarrow f_{\text{Adam}}(\Theta, \epsilon, \mathbf{G})$
  6: **end for**
  7: Output the trained model

---

(a)          (b)          (c)          (d)

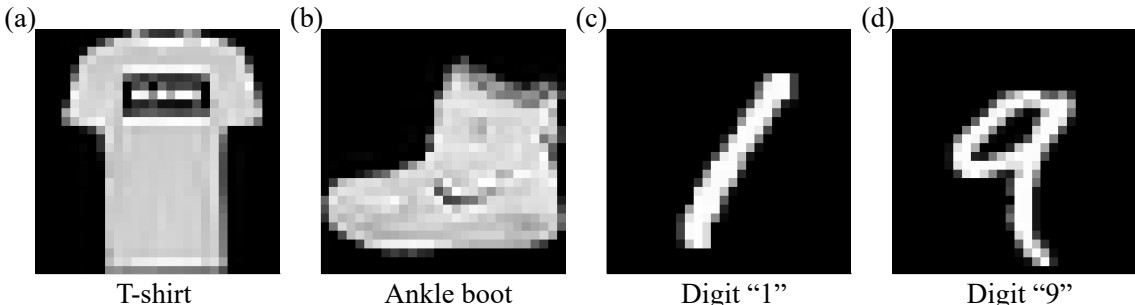

   T-shirt         Ankle boot        Digit "1"        Digit "9"

Figure 5: The visualization of the FashionMNIST and MNIST dataset, where both of them are originally 28 by 28 pixels. (a) A sample labeled "T-shirt" from the Fashion-MNIST dataset. (b) A sample labeled "Ankle boot" from the FashionMNIST dataset. (c) A handwritten digit "1" from the MNIST dataset. (d) A handwritten digit "9" from the MNIST dataset.

## 3.2 Benchmarks of the performance

Here, we provide numerical benchmarks of the amplitude-encoding based QNNs' performances with different hyper-parameters (depths, entangling layers) and four datasets: the FashionMNIST dataset [78], the MNIST handwritten digit dataset [79], an 8-qubit symmetry-protected topological (SPT) state dataset, and a 10-qubit symmetry-protected topological (SPT) state dataset. The FashionMNIST dataset and the MNIST handwritten digit dataset are well-known datasets that have been widely utilized in both commercial fields and scientific research. For the FashionMNIST dataset, we take the samples labeled "ankle boot" and "T-shirt" as our training and test data. For the MNIST handwritten digit dataset, we take the samples labeled "1" and "9" as our training and test data. In Fig. 5, we exhibit two samples for each of the above two classical datasets. For the SPT state dataset, we consider a one-dimensional cluster-Ising model with periodic boundary conditions with the following Hamiltonian [107]:

$$H(\lambda) = -\sum_{j=1}^{N} \hat{\sigma}_x^{(j-1)}\hat{\sigma}_z^{(j)}\hat{\sigma}_x^{(j+1)} + \lambda\sum_{j=1}^{N}\hat{\sigma}_y^{(j)}\hat{\sigma}_y^{(j+1)}, \tag{12}$$

where $\hat{\sigma}_\alpha^{(i)}$, $\alpha = x, y, z$, are Pauli matrices and $\lambda$ determines the relative strength of the nearest-neighbour interaction compared to the next-to-nearest-neighbor interaction. The model un-

dergoes a continuous quantum phase transition at $\lambda = 1$ which separates a cluster phase with a nonlocal hidden order for $\lambda < 1$ from an antiferromagnetic phase with long-range order and a nonvanishing staggered magnetization for $\lambda > 1$. We uniformly vary $\lambda$ from 0 to 2 at 0.001 intervals and obtain their corresponding ground states, which serve as our training set and test set.

The basic pseudocode for the learning procedure is shown in Algorithm 1. In the rest of this subsection, we will present detailed benchmarks considering the variables listed above, where the trained model's accuracy is the main criterion that will be exhibited in the following data tables. Moreover, the code for calculating the accuracy and loss over the training set, the test set, or a batch set is shown below:

```
using Quantum_Neural_Network_Classifiers: acc_loss_evaluation

# calculate the accuracy & loss for the training & test set
# with the method acc_loss_evaluation(circuit::ChainBlock,reg::ArrayReg,
# y_batch::Matrix{Float64},batch_size::Int64,pos_::Int64)
# pos_ denotes the index of the qubit to be measured
train_acc, train_loss = acc_loss_evaluation(circuit, x_train, y_train,
    num_train, pos_)
test_acc, test_loss = acc_loss_evaluation(circuit, x_test, y_test,
    num_test, pos_)
```

Table 2: Accuracy of the trained amplitude-encoding based QNN model with 12 different depths and 3 digital entanglement layers shown in Fig. 4(a-c). For each hyperparameter setting, the codes have been repeatedly run 100 times and the average test accuracy is exhibited.

| Datasets | FashionMNIST | | | MNIST | | | SPT(8 qubits) | | | SPT(10 qubits) | | |
|---|---|---|---|---|---|---|---|---|---|---|---|---|
| | $Ent_1$ | $Ent_2$ | $Ent_3$ | $Ent_1$ | $Ent_2$ | $Ent_3$ | $Ent_1$ | $Ent_2$ | $Ent_3$ | $Ent_1$ | $Ent_2$ | $Ent_3$ |
| **Block Depth** | | | | | | | | | | | | |
| 1 | 0.652 | 0.907 | 0.669 | 0.517 | 0.666 | 0.813 | 0.500 | 0.522 | 0.500 | 0.529 | 0.525 | 0.557 |
| 2 | 0.994 | 0.989 | 0.980 | 0.546 | 0.831 | 0.848 | 0.506 | 0.865 | 0.887 | 0.528 | 0.892 | 0.678 |
| 3 | 0.997 | 0.991 | 0.981 | 0.915 | 0.935 | 0.949 | 0.775 | 0.951 | 0.986 | 0.767 | 0.915 | 0.865 |
| 4 | 0.999 | 0.993 | 0.995 | 0.937 | 0.974 | 0.962 | 0.871 | 0.981 | 0.988 | 0.865 | 0.886 | 0.906 |
| 5 | 0.998 | 0.992 | 0.995 | 0.943 | 0.984 | 0.976 | 0.964 | 0.982 | 0.985 | 0.857 | 0.902 | 0.904 |
| 6 | 0.999 | 0.993 | 0.997 | 0.964 | 0.984 | 0.976 | 0.973 | 0.984 | 0.987 | 0.897 | 0.899 | 0.909 |
| 7 | 0.999 | 0.993 | 0.998 | 0.959 | 0.984 | 0.981 | 0.974 | 0.986 | 0.987 | 0.893 | 0.903 | 0.921 |
| 8 | 0.998 | 0.997 | 0.997 | 0.973 | 0.982 | 0.981 | 0.978 | 0.988 | 0.988 | 0.902 | 0.907 | 0.926 |
| 9 | 0.999 | 0.996 | 0.998 | 0.980 | 0.983 | 0.984 | 0.981 | 0.989 | 0.989 | 0.900 | 0.912 | 0.928 |
| 10 | 0.999 | 0.997 | 0.998 | 0.985 | 0.984 | 0.984 | 0.982 | 0.988 | 0.988 | 0.905 | 0.929 | 0.932 |
| 11 | 0.999 | 0.998 | 0.997 | 0.986 | 0.986 | 0.985 | 0.983 | 0.989 | 0.990 | 0.908 | 0.932 | 0.935 |
| 12 | 0.999 | 0.999 | 0.998 | 0.987 | 0.985 | 0.985 | 0.984 | 0.989 | 0.988 | 0.908 | 0.935 | 0.936 |

**a) Different depths with digital entangling layers.** In Table 2, we provide the numerical performances of the QNN classifiers introduced above with 12 different depths, 4 real-life and quantum datasets, and 3 different digital entangling layers shown in Fig. 4(a-c). We use the test accuracy as a measure of the trained model's performance. It should be noted that since different initial parameters may lead to very different training behaviors (very high accuracy at the beginning, stuck at local minima, volatile training curves, etc). To avoid being largely affected by occasional situations, we reinitialize the model 100 times and take the average test accuracy as the result. According to this table, we can see that the accuracy approximately increases with QNN classifiers' depths. We also find that when the QNN classifiers' depths are

relatively low (from 1 to 5), different digital entangling layers may lead to gapped performances. When the depths are over 10, the accuracy saturates to a decent value and the QNN classifiers with different entangling layers have similar performances, except for the case of learning 10-qubit SPT data.

Table 3: Accuracy of the trained amplitude-encoding based QNN model with 10 different Hamiltonian evolution time and 3 depths: $\text{Dep}_1 = 1$, $\text{Dep}_2 = 3$, and $\text{Dep}_3 = 5$. For each hyper-parameter setting, the codes have been repeatedly run 100 times and the average test accuracy is exhibited.

| Datasets | FashionMNIST | | | MNIST | | | SPT(8 qubits) | | | SPT(10 qubits) | | |
| --- | --- | --- | --- | --- | --- | --- | --- | --- | --- | --- | --- | --- |
| | $\text{Dep}_1$ | $\text{Dep}_2$ | $\text{Dep}_3$ | $\text{Dep}_1$ | $\text{Dep}_2$ | $\text{Dep}_3$ | $\text{Dep}_1$ | $\text{Dep}_2$ | $\text{Dep}_3$ | $\text{Dep}_1$ | $\text{Dep}_2$ | $\text{Dep}_3$ |
| **Time** | | | | | | | | | | | | |
| 0.1 | 0.977 | 0.994 | 0.996 | 0.577 | 0.707 | 0.918 | 0.579 | 0.810 | 0.883 | 0.631 | 0.671 | 0.758 |
| 0.3 | 0.995 | 0.996 | 0.998 | 0.777 | 0.960 | 0.980 | 0.720 | 0.961 | 0.978 | 0.678 | 0.864 | 0.906 |
| 0.5 | 0.999 | 0.996 | 0.998 | 0.906 | 0.979 | 0.988 | 0.862 | 0.979 | 0.984 | 0.728 | 0.894 | 0.909 |
| 0.7 | 0.967 | 0.996 | 0.996 | 0.901 | 0.982 | 0.985 | 0.931 | 0.985 | 0.986 | 0.804 | 0.907 | 0.907 |
| 1.0 | 0.969 | 0.991 | 0.996 | 0.857 | 0.985 | 0.983 | 0.961 | 0.983 | 0.987 | 0.865 | 0.905 | 0.918 |
| 2.0 | 0.910 | 0.992 | 0.996 | 0.954 | 0.986 | 0.984 | 0.948 | 0.986 | 0.987 | 0.903 | 0.913 | 0.920 |
| 3.0 | 0.901 | 0.991 | 0.996 | 0.944 | 0.985 | 0.985 | 0.909 | 0.981 | 0.987 | 0.861 | 0.921 | 0.908 |
| 5.0 | 0.993 | 0.997 | 0.998 | 0.947 | 0.983 | 0.984 | 0.933 | 0.975 | 0.985 | 0.914 | 0.915 | 0.923 |
| 7.0 | 0.931 | 0.992 | 0.996 | 0.966 | 0.981 | 0.984 | 0.968 | 0.982 | 0.988 | 0.889 | 0.916 | 0.919 |
| 10.0 | 0.994 | 0.996 | 0.997 | 0.962 | 0.982 | 0.985 | 0.979 | 0.986 | 0.988 | 0.911 | 0.908 | 0.923 |

**b) Analog layers' Hamiltonian evolution time.** In addition to the digital entangling layers, we also consider entangling layers that are implemented by a many-body Hamiltonian's time evolution as shown in Fig. 4(d). Once given a Hamiltonian, it is important to explore the relationship between the QNNs' performances and the Hamiltonian's evolution time. Here, we first fix the depth to be $\text{Dep}_1 = 1$, $\text{Dep}_2 = 3$, $\text{Dep}_3 = 5$ (if the QNN circuit is too deep, the accuracy will be very high regardless of the entangling layers) and set 10 discrete evolution time with the Aubry-André Hamiltonian [108]:

$$H/\hbar = -\frac{g}{2} \sum_k (\hat{\sigma}_x^{(k)} \hat{\sigma}_x^{(k+1)} + \hat{\sigma}_y^{(k)} \hat{\sigma}_y^{(k+1)}) - \sum_k \frac{V_k}{2} \hat{\sigma}_z^{(k)}. \tag{13}$$

Here, $g$ is the coupling strength, $V_k = V \cos(2\pi\alpha k + \phi)$ denotes the incommensurate potential where $V$ is the disorder magnitude, $\alpha = (\sqrt{5} - 1)/2$ and $\phi$ is randomly distributed on $[0, 2\pi)$ evenly. For simplicity, we set $g = 1$ and $V = 0$ before running each setting 100 times. According to the results from Table 3, it is shown that the performance will be relatively poor with a very short evolution time. This can be explained as, the shorter the evolution time, the closer the entangling layer will be to identity. Thus, these layers can not provide enough connections between different qubits. In addition, the accuracy may not have a positive relation with the evolution time at depth 1. For experimentalists, there might be more considerations for setting the evolution time, where the open-source codes may provide help.

**c) Different depths with analog entangling layers.** Similar to **a)**, here, we provide the numerical performances of the QNN classifiers with 12 different depths, 4 real-life and quantum datasets, and 3 different analog entanglement layers (evolution with the Aubry-André Hamiltonian with evolution time 5.0, 10.0, and 15.0). This is also complementary to the information provided in Table 3. As shown in Table 4, we find that the accuracy is relatively low at depth 1 for several cases, which might be caused by the fact that the information of the input states can not be fully captured by measuring a certain qubit (some qubits are outside the "light cone" of the measured qubit). When the depth increases, the accuracy quickly con-

Table 4: Accuracy of the trained amplitude-encoding based QNN model with 12 different depths and 3 analog entanglement layers. For each hyper-parameter setting, the codes have been repeatedly run 100 times and the average test accuracy is exhibited.

| Datasets | FashionMNIST | | | MNIST | | | SPT(8 qubits) | | | SPT(10 qubits) | | |
|---|---|---|---|---|---|---|---|---|---|---|---|---|
| | $Ent_1$ | $Ent_2$ | $Ent_3$ | $Ent_1$ | $Ent_2$ | $Ent_3$ | $Ent_1$ | $Ent_2$ | $Ent_3$ | $Ent_1$ | $Ent_2$ | $Ent_3$ |
| **Block Depth** | | | | | | | | | | | | |
| 1 | 0.989 | 0.991 | 0.900 | 0.927 | 0.915 | 0.894 | 0.914 | 0.968 | 0.949 | 0.904 | 0.901 | 0.864 |
| 2 | 0.992 | 0.989 | 0.993 | 0.960 | 0.944 | 0.967 | 0.958 | 0.974 | 0.972 | 0.909 | 0.908 | 0.865 |
| 3 | 0.995 | 0.993 | 0.993 | 0.975 | 0.974 | 0.975 | 0.960 | 0.982 | 0.979 | 0.906 | 0.911 | 0.861 |
| 4 | 0.996 | 0.994 | 0.995 | 0.979 | 0.979 | 0.977 | 0.976 | 0.984 | 0.983 | 0.906 | 0.907 | 0.894 |
| 5 | 0.997 | 0.995 | 0.995 | 0.982 | 0.981 | 0.979 | 0.980 | 0.985 | 0.985 | 0.918 | 0.917 | 0.898 |
| 6 | 0.998 | 0.996 | 0.996 | 0.981 | 0.982 | 0.980 | 0.983 | 0.986 | 0.986 | 0.924 | 0.923 | 0.914 |
| 7 | 0.996 | 0.996 | 0.997 | 0.982 | 0.983 | 0.982 | 0.985 | 0.987 | 0.986 | 0.923 | 0.927 | 0.925 |
| 8 | 0.997 | 0.996 | 0.997 | 0.983 | 0.982 | 0.982 | 0.988 | 0.987 | 0.987 | 0.928 | 0.930 | 0.923 |
| 9 | 0.997 | 0.996 | 0.997 | 0.983 | 0.984 | 0.984 | 0.987 | 0.987 | 0.987 | 0.931 | 0.931 | 0.926 |
| 10 | 0.998 | 0.997 | 0.997 | 0.985 | 0.984 | 0.985 | 0.988 | 0.988 | 0.988 | 0.930 | 0.931 | 0.932 |
| 11 | 0.997 | 0.997 | 0.997 | 0.985 | 0.985 | 0.985 | 0.987 | 0.988 | 0.988 | 0.934 | 0.934 | 0.931 |
| 12 | 0.997 | 0.998 | 0.997 | 0.987 | 0.986 | 0.986 | 0.988 | 0.989 | 0.989 | 0.932 | 0.934 | 0.933 |

verges to a high value, which is consistent with the results of the amplitude-encoding based QNNs utilizing digital entangling layers shown in Table 2.

# 4 Block-encoding based QNNs

## 4.1 The list of variables and caveats

In this section, we explore the performances of block-encoding based QNNs under different hyper-parameter settings. Similar to the section for amplitude-encoding based QNNs, here we list the hyper-parameters that will be changed to different values to test the corresponding performance. The exhibition of block-encoding based QNNs' performances will be left to the next subsection.

**Entangling layers.** The entangling layers are the same as Section 3, and we refer to that section for more details.

**A caveat on the scaling of encoded elements.** For block-encoding based QNNs, we need to encode the input data into the blocks, and the rotation angles of single-qubit gates are popular choices. However, given a data point vectorized as $\vec{x} = (x_1, x_2, \ldots, x_N)$, we need to decide the scaling factor $c$ such that each element $x_i$ will be encoded as $cx_i$ as a rotation angle. We find that the performance during the training process is very sensitive to this scaling factor, which should be considered seriously. We will numerically verify this property in the next subsection to provide some guidance for future works.

Here, we provide a framework for the data preparation and QNN circuit's design. The 10-qubit QNN circuit in our numerical simulations can be decomposed into nine composite blocks, where each one contains three layers of variational single-qubit gates and one layer of entangling gates as shown in Fig. 4(e). The number of available variational parameters is thus 270. When handling the 256-dimensional datasets, we choose to add 14 zeros at the end of the data vectors such that their dimensions can match.

```
using Yao, YaoPlots, MAT
using Quantum_Neural_Network_Classifiers: ent_cx, params_layer
```

```
# import the FashionMNIST data
vars = matread("../dataset/FashionMNIST_1_2_wk.mat")
num_qubit = 10

# set the size of the training set and the test set
num_train = 500
num_test = 100
# set the scaling factor for data encoding c = 2
c = 2
x_train = real(vars["x_train"][:,1:num_train])*c
y_train = vars["y_train"][1:num_train,:]
x_test = real(vars["x_test"][:,1:num_test])*c
y_test = vars["y_test"][1:num_test,:];

# define the QNN circuit, some functions have been defined before
depth = 9
circuit = chain(chain(num_qubit, params_layer(num_qubit),
        ent_cx(num_qubit)) for _ in 1:depth)
# assign random initial parameters to the circuit
dispatch!(circuit, :random)
# record the initial parameters
ini_params = parameters(circuit);
YaoPlots.plot(circuit)
```

For the $i$-th variational single-qubit gate, we encode $\theta_i + cx_i$ into its rotation angle to create an interleaved data encoding structure. To exhibit this idea, here we select and present an interleaved block-encoding QNN's encoding strategy in advance, whose performance will be tested in the next subsection with complete codes at Github QNN:

```
# the idea of block-encoding based QNNs through a simple example:
# the FashionMNIST dataset has been resized to be 256-dimensional
# we expand them to 270-dimensional by adding zeros at the end
dim = 270
x_train_ = zeros(Float64,(dim,num_train))
x_train_[1:256,:] = x_train
x_train = x_train_
x_test_ = zeros(Float64,(dim,num_test))
x_test_[1:256,:] = x_test
x_test = x_test_

# the input data and the variational parameters are interleaved
# this strategy has been applied to [Ren et al, Experimental quantum
# adversarial learning with programmable superconducting qubits,
# arXiv:2204.01738]. later we will numerically test the expressive
# power of this encoding strategy
train_cir = [chain(chain(num_qubit, params_layer(num_qubit),
        ent_cx(num_qubit)) for _ in 1:depth) for _ in 1:num_train]
test_cir = [chain(chain(num_qubit, params_layer(num_qubit),
        ent_cx(num_qubit)) for _ in 1:depth) for _ in 1:num_test];
for i in 1:num_train
    dispatch!(train_cir[i], x_train[:,i]+ini_params)
```

---

**Algorithm 2** Block-encoding based quantum neural network classifier

---

**Input:** The untrained model $h$ with variational parameters $\Theta$, the loss function $L$, the training
set $\{(\mathbf{x}_m, \mathbf{a}_m)\}_{m=1}^n$ with size $n$, the batch size $n_b$, the number of iterations $T$, the learning
rate $\epsilon$, and the Adam optimizer $f_{\text{Adam}}$

**Output:** The trained model

1: Initialization: generate random initial parameters for $\Theta$
2: **for** $i \in [T]$ **do**
3:     Randomly choose $n_b$ samples $\{\mathbf{x}_{(i,1)}, \mathbf{x}_{(i,2)}, ..., \mathbf{x}_{(i,n_b)}\}$ from the training set
4:     Calculate the gradients of $L$ with respect to the parameters $\Theta$, and take the average
    value over the training batch $\mathbf{G} \leftarrow \frac{1}{n_b}\Sigma_{k=1}^{n_b}\nabla L(h(\mathbf{x}_{(i,k)};\Theta), \mathbf{a}_{(i,k)})$
5:     Updates: $\Theta \leftarrow f_{\text{Adam}}(\Theta, \epsilon, \mathbf{G})$
6: **end for**
7: Output the trained model

---

```
end
for i in 1:num_test
    dispatch!(test_cir[i], x_test[:,i]+ini_params)
end
```

## 4.2 Benchmarks of the performance

Here, we provide numerical benchmarks of the block-encoding based QNNs' performances
with different hyper-parameters (scaling factors for data encoding, entangling layers) and
two datasets: the FashionMNIST dataset [78] and the MNIST handwritten digit dataset [79].
The basic pseudocode for the block-encoding based QNNs' learning procedure is shown in
Algorithm 2. In the rest of this subsection, we will benchmark the QNNs' performances with
respect to the variables listed above, where the achieved accuracy will be exhibited in the
following data tables. The code for calculating the accuracy and loss over the training set, the
test set, or a batch set is shown below:

```
using Quantum_Neural_Network_Classifiers: acc_loss_evaluation

# calculate the accuracy & loss for the training & test set
# with the method acc_loss_evaluation(nbit::Int64,circuit::Vector,
# y_batch::Matrix{Float64},batch_size::Int64,pos_::Int64)
# pos_ denotes the index of the qubit to be measured
train_acc, train_loss = acc_loss_evaluation(num_qubit, train_cir, y_train,
    num_train, pos_)
test_acc, test_loss = acc_loss_evaluation(num_qubit, test_cir, y_test,
    num_test, pos_)
```

    **a) Different scaling factors for data encoding with digital entangling layers.** In Table 5, we provide the numerical performances of the block-encoding based QNN classifiers
with 15 different scaling factors for data encoding, 2 real-life datasets, and 3 different digital
entangling layers shown in Fig. 4(a-c). Here, we mention that, before applying the scaling
factor for data encoding, the original data vectors are already normalized. Similar to the
amplitude-encoding based QNNs' setting, we reinitialize the model 100 times and take the
average test accuracy as the result. The numerical results in Table 5 reveal an important fact
that the optimal scaling factors for data encoding lie in a certain interval, while higher or

Table 5: Accuracy of the trained block-encoding based QNN model with 15 different scaling factors for data encoding and 3 digital entangling layers shown in Fig. 4(a-c). For each hyper-parameter setting, the codes have been repeatedly run 100 times and the average test accuracy is exhibited.

| Datasets | FashionMNIST | | | MNIST | | |
|---|---|---|---|---|---|---|
| | $Ent_1$ | $Ent_2$ | $Ent_3$ | $Ent_1$ | $Ent_2$ | $Ent_3$ |
| **Scaling Factor** | | | | | | |
| 0.1 | 0.526 | 0.531 | 0.537 | 0.579 | 0.574 | 0.583 |
| 0.4 | 0.776 | 0.786 | 0.765 | 0.731 | 0.768 | 0.736 |
| 0.7 | 0.959 | 0.964 | 0.972 | 0.880 | 0.863 | 0.874 |
| 1.0 | 0.988 | 0.988 | 0.988 | 0.940 | 0.948 | 0.945 |
| 1.3 | 0.991 | 0.992 | 0.991 | 0.965 | 0.967 | 0.965 |
| 1.6 | 0.994 | 0.993 | 0.992 | 0.972 | 0.975 | 0.973 |
| 2.0 | 0.994 | 0.994 | 0.994 | 0.976 | 0.976 | 0.975 |
| 2.4 | 0.995 | 0.994 | 0.994 | 0.978 | 0.978 | 0.977 |
| 2.8 | 0.995 | 0.994 | 0.994 | 0.977 | 0.976 | 0.979 |
| 3.2 | 0.995 | 0.995 | 0.994 | 0.976 | 0.975 | 0.977 |
| 4.0 | 0.993 | 0.992 | 0.994 | 0.967 | 0.965 | 0.966 |
| 5.0 | 0.987 | 0.987 | 0.988 | 0.938 | 0.938 | 0.931 |
| 6.0 | 0.979 | 0.978 | 0.980 | 0.878 | 0.874 | 0.882 |
| 8.0 | 0.947 | 0.949 | 0.947 | 0.740 | 0.736 | 0.747 |
| 10.0 | 0.895 | 0.896 | 0.889 | 0.644 | 0.641 | 0.637 |

lower of them may lead to comparably low performances. In our simulations for handling 256-dimensional data with a 10-qubit QNN circuit, choosing scaling factors between 1.5-3.5 results in a decent performance. It is worthwhile to mention that, in Ref. [61], we have experimentally demonstrated quantum adversarial machine learning for a 256-dimensional medical dataset with block-encoding based QNNs. The QNN structure utilized on the superconducting platform is similar to the ones in this paper, and we also carefully choose a scaling factor for better experimental performance.

**b) Analog layers' Hamiltonian evolution time.** Here, to explore the effect of analog layers' Hamiltonian evolution time on the block-encoding based QNNs' performance, we first fix the scaling factor for data encoding to be $Enc_1 = 1.5$, $Enc_2 = 2.0$, and $Enc_3 = 2.5$. Then, we set 10 discrete evolution time with the Aubry-André Hamiltonian. The model for each setting has been reinitialized and run 100 times to provide an average test accuracy. The numerical results are shown in Table 6, where we see that the evolution time turns out to have a minor influence on the average performance. In Table 3, we find that a low-depth QNN circuit may perform poorly with a short evolution time. This does not happen in the present block-encoding based QNNs' setting since the QNNs' depth in this setting is considerable and multiple entangling layers provide enough connections between different qubits.

## 5 Conclusion and outlooks

In this paper, we have briefly reviewed the recent advances in quantum neural networks and utilized Yao.jl's framework to construct our code repository which can efficiently simulate various popular quantum neural network structures. Moreover, we have carried out extensive numerical simulations to benchmark these QNNs' performances, which may provide helpful guidance for both developing powerful QNNs and experimentally implementing large-scale demonstrations of them.

Meanwhile, we mention that our open-source project is not complete since there are many

Table 6: Accuracy of the trained block-encoding based QNN model with 10 different Hamiltonian evolution time and 3 scaling factors for data encoding: $Enc_1 = 1.5$, $Enc_2 = 2.0$, and $Enc_3 = 2.5$. For each hyper-parameter setting, the codes have been repeatedly run 100 times and the average test accuracy is exhibited.

| Datasets | FashionMNIST | | | MNIST | | |
|---|---|---|---|---|---|---|
| | $Enc_1$ | $Enc_2$ | $Enc_3$ | $Enc_1$ | $Enc_2$ | $Enc_3$ |
| Time | | | | | | |
| 0.1 | 0.992 | 0.994 | 0.995 | 0.972 | 0.974 | 0.979 |
| 0.3 | 0.992 | 0.994 | 0.994 | 0.970 | 0.978 | 0.977 |
| 0.5 | 0.992 | 0.994 | 0.995 | 0.969 | 0.978 | 0.978 |
| 0.7 | 0.991 | 0.994 | 0.994 | 0.969 | 0.979 | 0.978 |
| 1.0 | 0.992 | 0.995 | 0.995 | 0.972 | 0.976 | 0.977 |
| 2.0 | 0.993 | 0.994 | 0.995 | 0.970 | 0.977 | 0.979 |
| 3.0 | 0.993 | 0.994 | 0.995 | 0.974 | 0.977 | 0.976 |
| 5.0 | 0.993 | 0.993 | 0.994 | 0.973 | 0.977 | 0.979 |
| 7.0 | 0.991 | 0.994 | 0.995 | 0.970 | 0.977 | 0.979 |
| 10.0 | 0.992 | 0.994 | 0.995 | 0.971 | 0.979 | 0.979 |

interesting structures and training strategies of QNNs unexplored by us, e.g. quantum convolutional neural networks, quantum recurrent neural networks, etc. We would be very happy to communicate with members from both the classical and quantum machine learning communities to discuss about and enrich this project.

For the future works, we would like to mention several points that may trigger further explorations:

- At the current stage, the non-linearity of quantum neural networks can not be designed flexibly, since the QNN circuit itself can be seen as a linear unitary transformation. This seems like a restriction compared with classical neural networks, where the latter can flexibly design and utilize non-linear transformations such as activation functions, pooling layers, convolutional layers, etc. A possible future direction is to explore how to efficiently implements flexible non-linear transformations to enhance QNNs' expressive power, or to design hybrid quantum-classical models such that the quantum and classical parts of the model can demonstrate their advantages, respectively.

- As mentioned in [51, 96], the data-encoding strategies are important for the performance. In our numerical simulations, we mainly adjust the hyper-parameters to empirically improve QNNs' performance. More theoretical guidance will be helpful for future works, especially for near-term and large-scale experimental demonstrations.

- In this work, the variational parameters are mainly encoded into the single-qubit rotation gates. One may also explore the possibility of encoding the parameters into some global operations. For example, considering the time evolution of Aubry-André Hamiltonian, the evolution time and coupling strength can be utilized as variational parameters to be optimized. For numerical simulations, we can use Yao.jl to define such customized gates and utilize the builtin automatic differentiation (AD) engine or a general AD engine such as Zygote.jl [109] to efficiently simulate the optimization procedure.

- Interpretable quantum machine learning: in recent years, the interpretability of machine learning models has been considered crucially important, especially in sensitive applications such as medical diagnosis or self-driving cars [110]. Along this line, popular methods used in interpreting deep learning models include saliency maps and occlusion

maps, which explore the importance of one pixel in an image to the final evaluation function. To our knowledge, there are few works focusing on this direction [111], and further explorations might be needed.

# Acknowledgements

We would like to thank Wenjie Jiang, Weiyuan Gong, Xiuzhe Luo, Peixin Shen, Zidu Liu, Sirui Lu, anonymous reviewers for helpful discussions and comments, and Jinguo Liu in particular for very helpful communications about the organization of the code repository. We would also like to thank the developers of Yao.jl for providing an efficient quantum simulation framework and Github for the online resources to help open source the code repository.

**Funding information** This work is supported by the start-up fund from Tsinghua University (Grant. No. 53330300322), the National Natural Science Foundation of China (Grant. No. 12075128), and the Shanghai Qi Zhi Institute.

# A Preparation before running the code

The quantum neural networks in this work are built under the framework of Yao.jl in Julia Programming Language. Detailed installation instructions of Julia and Yao.jl can be found at Julia and Yao.jl, respectively.

The environments for the codes provided in jupyter-notebook formats can be built with the following commands:

```
$ git clone https://github.com/LWKJJONAK/Quantum_Neural_Network_Classifiers
$ cd Quantum_Neural_Network_Classifiers
$ julia --project=amplitude_encode -e "using Pkg; Pkg.instantiate()"
$ julia --project=block_encode -e "using Pkg; Pkg.instantiate()"
```

In addition, for better compatibility, using version 1.7 or higher of Julia is suggested.

# B Complete example codes

The codes for numerical simulations to benchmark QNNs' performances can be found at:
    https://github.com/LWKJJONAK/Quantum_Neural_Network_Classifiers,
where we provide:

- Detailed tutorial codes for building amplitude-encoding based and block-encoding based QNNs with annotations.

- All the data generated for the tables in this paper from Table 2 to Table 6, which includes more than 55000 files in ".mat" format. In addition to the average accuracy provided in the paper, in the complete data files, we also record the learning rate, the batch size, the number of iterations, the size of the training and test sets, and the accuracy/loss curves during the training process.

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
