# Peer review of "Quantum Neural Network Classifiers: A Tutorial"

_SciPost Physics Lecture Notes, doi:SciPost Phys. Lect. Notes 61 (2022)_

## Round 1 · Referee Report · Anonymous (Referee 1) · 2022-7-3

Report

The manuscript focuses on providing a tutorial on quantum neural networks implemented by parameterized quantum circuits. The source code realized by Julia is available. The submission is generally well written and structured. I will be happy to recommend publication if the following issues are well addressed.

• In the Introduction, the motivation for investigating quantum neural networks in the form of variational quantum circuits should be further explained from both the theoretical and experimental sides. At the current version, it seems unclear why we need to study quantum neural networks since deep neural networks are sufficiently powerful.

• In Section 2, the introduction of quantum computing should be appended. The current version is unfriendly for the readers coming from the computer science community.

• In Section 3, it is better to explain and visualize the employed datasets (e.g., MNIST, Fashion MNIST) for ease of understanding.

• Although it is unnecessary to present all quantum classifiers in the tutorial, the readability of the tutorial can be enhanced if the authors can provide a categorization of these quantum classifiers somewhere.

• The notation about the gradients is mismatched. In Algorithms 1&2, the symbol L and \mathcal{L} are mixedly used.

• Page 15, ‘have a minor influence one’--> ‘have a minor influence on'.
  • validity: -
  • significance: -
  • originality: -
  • clarity: -
  • formatting: -
  • grammar: -

Author:  Weikang Li  on 2022-07-05  [id 2631]

(in reply to Report 1 on 2022-07-03)
Category:
answer to question

We thank the referee for his/her/their time in reviewing the manuscript and for his/her/their kind recommendation of our paper. The detailed point-by-point response to the referee’s comment/suggestion is provided below.

Q1: We agree with the referee that the motivation for developing QNN classifiers should be explained in depth. As a modification, we add discussions about this point from both the theoretical and experimental sides in the Introduction.

From the theoretical side, we explain the motivation in the third paragraph of the Introduction as the following:

  1. Motivated by the success of classical deep learning as well as advances in quantum computing, quantum neural networks (QNNs), which share similarities with classical neural networks and contain variational parameters, have drawn a wide range of attention. There are multiple reasons to develop the quantum version of neural networks:
  2. First, quantum computers hold the potential to outperform classical computers from several aspects: Some quantum Fourier transform based algorithms, such as Shor's factoring algorithm \cite{Shor1997PolynomialTime}, can achieve exponential speedups compared with the best known classical methods; Moreover, some quantum resources such as quantum nonlocality and contextuality are proved to offer unconditional quantum advantages in solving certain computational problems \cite{Bravyi2018Quantum,Bravyi2020Quantum}. These fascinating results stimulate the exploration of potential advantages in QNN models, especially in the age of big data.
  3. Second, when we are trying to learn from a quantum dataset, i.e., a dataset whose data is generated from a quantum process, instead of from a classical dataset, it would be more natural to use a quantum model the handle the task: Extracting enough information from a quantum state to a classical device would be very challenging when the system scales up, while a QNN model which can handle the data in the exponential-large Hilbert space naturally may provide certain advantages. This point has been explicitly demonstrated in Ref.~\cite{Cong2019Quantum}, where the QNN model proposed in this work can recognize quantum states of one-dimensional symmetry-protected topological phases better than existing approaches.
  4. Third, from the aspect of effective dimension, which is a property related to a model’s generalization performance on new data, there is preliminary evidence showing that quantum neural networks may be able to achieve better effective dimension and faster training than comparable feedforward networks \cite{Abbas2021Power}.

    From the experimental side, we choose to introduce the experimental advances over the past 5 years and explain that the experimental demonstration has evolved from small proof-of-principle experiments to large-scale and high-dimensional ones, which shows that the QNN classifiers on real devices are also growing rapidly towards real-life applications. This is explained in the fourth paragraph of the Introduction as the following:

  5. For the experimental demonstrations, several QNN models have been implemented experimentally. In Ref.~\cite{Grant2018Hierarchical}, a proof-of-principle QNN classifier is deployed on the ibmqx4 quantum computer to classify the Iris data. In Ref.~\cite{Havlicek2019Supervised}, a QNN classifier is utilized to train and classify some artificially-generated samples on a superconducting quantum processor. With the rapid development of quantum devices, more recently, QNN classifiers are utilized to handle high-dimensional real-life datasets or quantum datasets. In Ref.~\cite{Herrmann2021Realizing}, the authors experimentally demonstrated a quantum convolutional neural network model to identify symmetry-protected topological phases of a spin model on 7-qubit superconducting platforms. In Ref.~\cite{Gong2022Quantum}, a quantum neuronal sensing model has been proposed to classify ergodic and localized phases of matter with a 61-qubits superconducting quantum processor. Moreover, in Ref.~\cite{Ren2022Experimental}, two QNN classifiers have been experimentally demonstrated for classifying classical and quantum datasets, respectively, where an interleaved QNN classifier is trained on a 36-qubit quantum processor (10 qubits are used) experimentally which turns out to accurately classify 256-dimensional medical images and handwritten digits, and another 10-qubit QNN classifier successfully classifies the quantum states generated by evolving the N\'{e}el state with the Aubry-Andr\'{e} Hamiltonian. In addition to the fact that QNNs are candidates to be commercially available in the noisy intermediate-scale quantum (NISQ) era, it is an interesting point to see that these experimental QNN classifiers introduced above are also suitable to be a measure of progress in quantum techniques during this era.

Q2 & Q4: We agree that the original version without an introduction of quantum computing and quantum classifiers might be less convenient for readers from other communities. In the current version, we add a new subsection titled “A recap of quantum computing and quantum classifiers”, including “The basic knowledge of quantum computing” and “A categorization of quantum classifiers”.

In “The basic knowledge of quantum computing”, we start by introducing the history of quantum computing (the proposal of quantum Turing machines, Feynman’s famous talk about simulating physics with quantum computers, the influence of Shor’s algorithm, etc). Then, we introduce the basic units in quantum computing (qubits) and some basic properties such as superposition and entanglement. Furthermore, we have mentioned several important quantum algorithms (1. Quantum algorithms related to quantum Fourier transform; 2. Grover’s algorithm; 3. Bravyi’s works about unconditional separations between quantum and classical circuits) and introduced their advantages over their classical counterparts.

In “A categorization of quantum classifiers”, we first introduce the basic definition and concepts of quantum classifiers. Then, we present a broader range of quantum classifiers beyond QNN classifiers such as quantum nearest-neighbor algorithms, quantum decision tree classifiers, quantum kernel methods, and quantum support vector machines. These categories are summarized in a new Table with representative works exhibited and briefly introduced.

Q3: We thank the referee for this suggestion. We add a new figure to exhibit the visualization of the MNIST and FashionMNIST datasets in the revised manuscript.

Q5 & Q6: We thank the referee for his/her/their careful check of our manuscript, and these two typos have been fixed in the revised manuscript.

In summary, we have added more information to this manuscript to make this tutorial more complete, including the motivations of QNN classifiers, an introduction to quantum computing and quantum classifiers, a new table, and a new figure. We also changed the style of the code box for better readability in both the PDF and printed versions.

---

## Round 1 · Referee Report · Anonymous (Referee 1) · 2022-7-7

Report

The authors' response well addressed my concerns. I am happy to recommend the submission for publication.

---

## Round 1 · Referee Report · Anonymous (Referee 2) · 2022-7-9

Report

The manuscript provides a introductory tutorial into quantum machine learning with variational quantum circuits. The authors provide a step-by-step explanation of quantum machine learning and provide code based on Yao in Julia to train quantum models. This tutorial is well written and of high value for researchers starting in the field. My concern is that some aspects are not sufficiently explained in the current manuscript.

I recommend publication after my comments below are addressed:

  • in introduction: Alone this line -->Along this line
  • 2.1.2 V_i(\theta_j): I believe the index should match
  • in the discussion of the shift-rule, it would be good to explain why the shift-rule is better than other methods such as finite-differences
  • While the authors mention the parameter shift rule, in the actual code the gradients are computed with a different method, namely automatic differentiation as implemented by Yao. The authors should mention this fact.
  • it should be explained more clearly how parameters are mapped to quantum gates in practice, i.e. how many parameters can be encoded per qubit/layer? Which gates are best to choose? Is data re-uploading used, or each parameter is mapped to a single gate?
  • For block-encoding, it may be helpful for readers to discuss how the authors choose the number of qubits/layers for their examples.
  • The authors should state more clearly how they measured the cost function (i.e. was \sigma^z measured on a single qubit?)
  • Why was Eq.9 Aubry-Andre chosen as entangling Hamiltonian, when then V was set to zero? This case is in fact simply the well known XY model
  • When using a quantum computer for machine learning, expectation values cannot be evaluated exactly, but are estimated with finite accuracy using a finite number of measurements. The authors should discuss this aspect and how it affects training and the cost function.
  • validity: -
  • significance: -
  • originality: -
  • clarity: -
  • formatting: -
  • grammar: -

Author:  Weikang Li  on 2022-07-13  [id 2657]

(in reply to Report 3 on 2022-07-09)

We thank the referee for his/her/their time in reviewing the manuscript and for his/her/their kind recommendation of our paper. The detailed point-by-point response to the referee’s comments/questions is provided below.

Q1&Q2: We thank the referee for pointing out these two typos. In the revised version, these typos as well as several other ones will be fixed.

Q3: We thank the referee for this suggestion. Indeed, it will be clearer to analyze the advantages of parameter shift rule over methods like finite difference. We add the following explanations for this purpose:

1. When we look at the expression of the parameter shift rule, we may connect it to a widely-applied approximation method called the finite difference method. In our case, the above derivative can be approximately expressed as [expression]. In addition to the fact that the finite difference method is not exact, the unavoidable experimental noise will affect the result with the finite difference method more than that with the parameter shift rule, thus this approximation method is less practical.

Q4: Yes, in numerical simulations, we applied the automatic differentiation rather than the parameter shift rule. The reason is that we wish to boost the running speed of the QNNs’ simulation, while the automatic differentiation in our setting is about 30 times faster than “simulating” the parameter shift rule. This does not mean that the parameter shift rule is not efficient, since the parameter shift rule is expected to run naturally on a real quantum computer rather than in numerical simulations. In the revised version of our manuscript, we add the following explanations:

1. In the codes attached in this paper, the gradients are calculated by automatic differentiation implemented by Yao.jl. The reason why we do not apply the parameter shift rule is that we wish to make the numerical simulations faster. Automatic differentiation fulfills this goal and meanwhile has analytical precision, thus being the one adapted in our work. When we have a real quantum computer to deploy large-scale quantum neural networks that are hard for classical computers to simulate, methods like the parameter shift rule would be a natural and favorable choice.

Q5&Q6: We thank the referee for pointing out that some settings implemented in the codes are not described in the manuscript. The rotation angle in a single-qubit rotation gate can be utilized to encode one variational parameter and data re-uploading is not used. Then it is clear that the number of parameters encoded per qubit/single-qubit-layer is the number of single-qubit-layers/qubits. For the block-encoding, we also add the following descriptions about some hyper-parameter settings:

1. The 10-qubit QNN circuit in our numerical simulations can be decomposed into nine composite blocks, where each one contains three layers of variational single-qubit gates and one layer of entangling gates as shown in Fig.4(e). The number of available variational parameters is thus 270. When handling the 256-dimensional datasets, we choose to add 14 zeros at the end of the data vectors such that their dimensions can match. For the i-th variational single-qubit gate, we encode $\theta_i + c x_i$ into its rotation angle to create an interleaved data encoding structure.

Q7: Yes, the $\sigma_z$ is measured on a single qubit since we mainly focus on binary classifications in this tutorial. In Sec 2.3.1, we add more descriptions for this point:

1. Here, g can be obtained by measuring some qubits on the Z-basis. In our tutorial which focuses on binary classifications, we choose to repeatedly measure one qubit on the Z-basis to evaluate g = (g1, g2), where the choice of this qubit's index can be flexible.

Q8: Yes, when V was set to zero, it becomes the XY model. Here, we aim to use a simple example for demonstrations and leave the complete code in the Github repository so that the readers can change these hyper-parameters according to their needs.

Q9: We thank the referee for pointing out the need to analyze the errors caused by a finite number of measurements. To better address this issue, we add a new subsection titled “Effects of finite measurements and experimental noises”, where we discussed both the complexity of repeat measurements and the influence of experimental noises:

1. In the above discussions about the optimization procedure, the calculation of gradients is closely related to some expectation values obtained from quantum measurements. However, unlike the numerical simulations where we can calculate the accurate expectation values, we can only apply a finite number of measurements to approximate these values with a real quantum computer. Moreover, the unavoidable experimental noises will further drive the outputs away from the accurate values.
2. First, for the effects of a finite number of measurements, there are a constant number of discrete measurement outcomes with different probabilities. According to the Chernoff bound, $O(\frac{1}{\epsilon^2})$ repeat measurements are needed to achieve the evaluation of an expectation value with an additive error up to $\epsilon$. At the current stage, the superconducting quantum processors can accomplish thousands of single-qubit measurements in several seconds, which is suitable for demonstrating QNNs' learning process \cite{Ren2022Experimental,Gong2022Quantum,Herrmann2021Realizing}. Second, since the experimental noises make the evaluations less accurate, the QNN model may converge slower and even get stuck into barren plateaus \cite{Wang2021NoiseInduced}, which might be intractable when the system scales up. To handle this issue, on the one hand, it is necessary to push the experimental limit and improve the gate fidelities. On the other hand, developing QNN models with higher noise robustness is of crucial importance.

---

## Round 2 · List of Changes

1. Following the referee’s suggestion, we added more discussions about the motivations for developing QNN classifiers in the Introduction section.
2. Minor revisions are made (mostly about incorrectly used words).
3. We added a new subsection titled “A recap of quantum computing and quantum classifiers”, including “The basic knowledge of quantum computing” and “A categorization of quantum classifiers”, aiming to provide convenience for readers unfamiliar with this field.
4. We changed the style of the code box for better readability in both the PDF and printed versions.
5. More explanations for some basic concepts.
6. We added a new subsection titled “Effects of finite measurements and experimental noises”.

---

## Editorial Decision

published